# A Population Dynamic Model to Assess the Diabetes Screening and Reporting Programs and Project the Burden of Undiagnosed Diabetes in Thailand

**DOI:** 10.3390/ijerph16122207

**Published:** 2019-06-21

**Authors:** Wiriya Mahikul, Lisa J White, Kittiyod Poovorawan, Ngamphol Soonthornworasiri, Pataporn Sukontamarn, Phetsavanh Chanthavilay, Wirichada Pan-ngum, Graham F Medley

**Affiliations:** 1Department of Tropical Hygiene, Faculty of Tropical Medicine, Mahidol University, Bangkok 10400, Thailand; wiriya_aon@hotmail.com (W.M.); ngamphol.soo@mahidol.ac.th (N.S.); 2Mahidol-Oxford Tropical Medicine Research Unit, Faculty of Tropical Medicine, Mahidol University, Bangkok 10400, Thailand; lisa@tropmedres.ac (L.J.W.); phetsavanh456@gmail.com (P.C.); 3Centre for Tropical Medicine, Nuffield Department of Medicine, University of Oxford, Oxford OX3 7BN, UK; 4Department of Clinical Tropical Medicine, Faculty of Tropical Medicine, Mahidol University, Bangkok 10400, Thailand; kittiyod.poo@mahidol.ac.th; 5College of Population Studies, Chulalongkorn University, Bangkok 10330, Thailand; pataporn@hotmail.com; 6Institute of Research and Education Development, UHS, Vientiane 7444, Laos; 7Centre for Mathematical Modelling of Infectious Disease & Department of Global Health and Development, London School of Hygiene and Tropical Medicine, Keppel Street, London WC1E 7HT, UK; Graham.Medley@lshtm.ac.uk

**Keywords:** population dynamic model, diabetes, undiagnosed diabetes, aging population, screening, reporting, mortality, Bayesian MCMC

## Abstract

Diabetes mellitus (DM) is rising worldwide, exacerbated by aging populations. We estimated and predicted the diabetes burden and mortality due to undiagnosed diabetes together with screening program efficacy and reporting completeness in Thailand, in the context of demographic changes. An age and sex structured dynamic model including demographic and diagnostic processes was constructed. The model was validated using a Bayesian Markov Chain Monte Carlo (MCMC) approach. The prevalence of DM was predicted to increase from 6.5% (95% credible interval: 6.3–6.7%) in 2015 to 10.69% (10.4–11.0%) in 2035, with the largest increase (72%) among 60 years or older. Out of the total DM cases in 2015, the percentage of undiagnosed DM cases was 18.2% (17.4–18.9%), with males higher than females (*p*-value < 0.01). The highest group with undiagnosed DM was those aged less than 39 years old, 74.2% (73.7–74.7%). The mortality of undiagnosed DM was ten-fold greater than the mortality of those with diagnosed DM. The estimated coverage of diabetes positive screening programs was ten-fold greater for elderly compared to young. The positive screening rate among females was estimated to be significantly higher than those in males. Of the diagnoses, 87.4% (87.0–87.8%) were reported. Targeting screening programs and good reporting systems will be essential to reduce the burden of disease.

## 1. Introduction

The prevalence of diabetes mellitus (DM) including type 1, type 2, and gestational diabetes is increasing globally and predicted to rise from 425 million adult cases in 2017 to 629 million in 2045 [1]. In Thailand, the increase of DM is more dramatic: The number of patients with physician-diagnosed diabetes and the intake of hypoglycemic drugs, rose from 0.4 million in 2005 to 2.3 million in 2015. DM incidence per 100,000 population rose from 48 in 2005 to 293 in 2015 (males) and 100 in 2005 to 414 in 2015 (females) [2]. About 70% of the population aged more than 60 years old in 2014 had been diagnosed for DM, and 20% of the population aged between 15 and 39 years [3,4]. The case fatality rate of diagnosed diabetes was about 18.4/1000 in 2006 [5]. Deaths from undiagnosed diabetes is unknown. Acute complications of diabetes include diabetic ketoacidosis (DKA), Euglycemic DKA, hyperosmolar hyperglycemic state and hypoglycemia [6,7]. Chronic complications of diabetes include microvascular complications such as neuropathy, retinopathy and nephropathy, and macrovascular complications such as coronary artery disease, peripheral arterial disease, and stroke [7,8].

Thailand is undergoing the first demographic and epidemiological transitions, so that the life expectancy may increase. The projection of disease assuming a static population would result in inaccurate figures [9], especially for diseases with age-related incidence [10]. Previous studies in the U.S. and Morocco showed diabetes trends were influenced by demographic changes [11,12]. Assessment of diabetes prevalence in the population is essential for public health planning, including prevention programs and medical needs provision [13]. 

The potential policies for diabetes management in Thailand include increasing the coverage of screening programs, and targeting screening among high risk groups, both of which facilitate early diagnosis and successful intervention to reduce and prevent the onset and progression of diabetes complications. These require explicit public health policies, such as, the improvement of reporting of both medical and economic burden, and health education for the prevention and control of diabetes [14]. In 2005, the Department of Disease Control formed the strategy linked to the strategic plans of the Ministry of Public Health (MoPH), which included improving the quality of diabetes screening programs and the development of the national screening service (see Appendix A). In 2005, at least 60% of the Thai population aged 40 years old or over would be asked to join the screening program recommended by the MoPH. In 2008, strategic plans were expanded to screen for diabetes among those aged over 34 years to at least 65%. In 2011, the target was raised to 90% per year. From 2013 to date, the target has been set to screen from the age of 15 years old with the coverage up to 90% [15,16].

The Bureau of Epidemiology (BoE) collaborated with The Office of Disease Prevention and Control to develop and launch the Chronic Disease Surveillance (CDS) system in April 2003. The disease burden and trends of Diabetes, Hypertension, and Ischemic heart disease were the main focus of this system. The CDS was applied nationally through all health service units: regional hospitals, general hospitals, community hospitals, health centers, and private hospitals. Data was sent via the CDS program to provincial health offices and then reported annually in parallel to the Bureau of Epidemiology, Ministry of Public Health, and the Regional Office of Disease Prevention and Control. In 2005, only 28/76 provinces were reported to the Provincial Health Office. From 2012 to date, data was collected from 76 provinces, except Bangkok where data would be sent directly to the Bureau of Policy and Strategy, Ministry of Public Health, and analyzed by the Bureau of Epidemiology, and Ministry of Public Health (see Appendix A) [2]. 

Many methods have been applied to project the diabetes burden [12,17,18], including mathematical modelling, which was used to predict and estimate disease burden [19,20,21,22] and help understand the role of risk factors [23]. It was predicted previously that the total diabetes population in Thailand would be approximately 5.4 million people in 2030 [17]. Some diabetes models assessed glucose and insulin dynamics, while relatively few models explained the epidemiology of diabetes [24]. Two studies measured population-based screening and the under-reporting of inpatient diabetes [25,26]. The main objective of this study is to use a population dynamic model overlaid with a diabetes dynamic sub-model to predict the disease burden, and in particular, the mortality of undiagnosed diabetes in the Thai population by age and sex, and to assess the diabetes screening program and reporting system. We used a Bayesian framework which permits the inclusion of uncertainty in onset and reporting rates.

## 2. Materials and Methods

### 2.1. Demographic Sub-Model

We generated the demographic deterministic sub-model as follows (see Appendix A). We divided the population by sex and age into 101 annual interval classes from 0 to 100 years old, where the population in each age class followed the actual population structure of Thailand between 1980 and 2015 [27,28], by using the birth, death, migration rates, and the 1980 census as the initial condition. All females in the age classes 15 to 50 years old can reproduce with the fertility rate (fr) [29], and the death rate is age-related [30]. The population older than 100 years were assumed to die. Crude net migration rate (immigrant minus emigrant per 1000) during each year was shared to all sex and age compartments [31]. The sex and age populations were obtained from the Population and Housing Census [28]. We solved a large set of Ordinary Differential Equations (ODE) of the demographic deterministic sub-model defined in the Appendix A.

### 2.2. Diabetes Dynamic Sub-Model

The demographic-sub model was then overlaid with the diabetes dynamic sub-model defined in the Appendix A. In the diabetes dynamic sub-model, type 1 and type 2 diabetes were not distinguished although globally 90% of all cases were type 2. Population were further divided into the three health statuses: healthy (*C^H^*), undiagnosed diabetes (*C^DM_un^*), and diagnosed diabetes (*C^DM^*) (Figure 1).

The diagnosed diabetes rates were gained from National Health Examination Surveys between 2005 and 2015 [3]. Diagnosed diabetes cases by age-group and sex were obtained from the annual epidemiological surveillance report between 2005 and 2015 [2]. Key assumptions for our model were as follows. First, the disease progression was irreversible, i.e., people cannot move from diabetic to non-diabetic. Second, we did not consider pre-diabetes or impaired glucose tolerance. Third, we assumed that incidence rates are constant over time but vary by age. It is possible that the incidence of diabetes could be influenced by some possible confounders such as behavior changes including lifestyle, diet, and exercise [32,33]. For the study’s simplicity, these factors were not included in the modelling.

We used R software version 3.2.3 (R Core Team, Vienna, Austria) to run and analyze the model outputs, and the deSolve package to solve differential equations [34]. The initial parameter values were calculated from population data and disease burden. Model fitting was carried out using the Markov Chain Monte Carlo (MCMC), implemented in the Bayesian Tools R package defined in the Appendix A [35]. Although the demographic sub-model was run from 1980, it was not until 2000 that diabetes was introduced into the model. The model was then run and fitted to the annual incidence and prevalence from 2005 to 2015. Six separate chains, each consisting of 35,000 iterations and a burn-in period of 5000 iterations were run in parallel to achieve a target acceptance rate of 0.15. The incidence rates, mortality of undiagnosed diabetes, and screening and reporting were estimated. The median value correlations between parameters and the credibility interval were reported. The model was further used to project 20-year age-specific prevalence and incidence of diabetes among males and females in Thailand, sampling all 32 parameters from the posterior chains. 

## 3. Results

The demographic sub-model was able to reproduce the population size of Thailand from 1980 to 2015 (see Appendix A) and the observed DM data with age and specific gender (see Appendix A). Through model fitting, the model parameters were estimated with the posterior distribution obtained (see Appendix A). Projections of the number of people with diabetes are given in Table 1. The largest increment in the number of people with diabetes was projected to occur in the eldest age category. 

Total DM burden was projected to increase by 34.1% by 2035, from 3.7 (3.6–3.8) million in 2015 to 4.9 (4.8–5.1) million in 2035. Overall, DM prevalence was projected to increase from 6.5% (6.3–6.7%) in 2015 to 10.7% (10.4–11.0%) in 2035. The number of men who were ≥60 years of age with DM would increase from 0.7 (0.68–0.72) million in 2015 to >1.3 (1.2–1.32) million in 2035 (+86%). While, the number of women with DM in this same age-group would rise from 1.1 (1.0–1.11) million in 2015 to >1.8 (1.72–1.85) million in 2035 (+64%). Age specific prevalence of DM in males and females was similar as shown in Figure 2 and Appendix A. 

Estimates of diabetes reporting’s proportion for each 5-year interval are given in Appendix A. The percentage of reported diabetes was increasing overtime: 87.4% (87.0–87.8%) in 2015. The positive screening rates among males increased over age groups in both periods, 2005–2009 and 2010–2015. Positive screening rates for males aged over 60 were ten-fold greater than among those aged between 15 and 34, about 1.11 vs. 0.06. From being undiagnosed to screening-positive among over 60-year-olds was around 1 year, whereas it would take over 10 years for aged between 15 and 34 years old. The positive screening rates among females were in the opposite trend, i.e., the screening rate was much higher among those aged 15 to 34 years old, with overall rate slightly greater in all groups when compared with males (see Appendix A). 

Estimates of undiagnosed diabetes case fatality rates for each 10-year interval are given in Appendix A. They were ten-fold higher than diagnosed diabetes. The largest fatality rates were among undiagnosed diabetes cases for ages between 15 and 39, about 1.5 (1–2 per person per 6 month). The estimates of DM incidence rates among females were higher than males except for those aged between 0 and 39 years old. The largest DM incidence rate was among females aged between 50 and 59 years old, about 330 (320–340 per 10,000 persons per year), as shown in Appendix A. 

Estimates of the prevalence of undiagnosed diabetes cases among females and males are given in Figure 3. Total undiagnosed diabetes was estimated to increase from 592,000 (589,000–598,000) in 2005 to 673,000 (664,000–684,000) in 2015 (18.2% of total diabetes cases). While, undiagnosed DM among males was higher and rose faster than females as shown in Appendix A.

The posterior distribution captures some significant pairs of correlation between parameters, i.e., the incidence rates by age were positively correlated with each other, so were the screening rates by age. The screening was negatively correlated with diabetes incidence rates and the incidence rates were positively correlated with the mortality of undiagnosed diabetes. There was no correlation between reporting and the other parameters. By using the MCMC approach, sampling from the posterior distribution had directly accounted for these correlations.

## 4. Discussion

Many statistical models have been used traditionally to predict incidence of diabetes on national and global scales. We applied the population dynamics with age and sex components and the health screening scheme to study diabetes in Thailand. Some similarities and differences of our findings when compared with some published work are discussed here. Whiting et al. took the logistic regression model together with the National Health Examination Surveys to predict the global estimates of DM prevalence to 2030 based on a linear prevalence trend [17]. They projected a 35% increase in the number of people with diabetes from 2011 (4.01 million) through to 2030 (5.45 million). Conversely, our finding showed a 57% increase of diabetes cases between 2011 (3.27 million) and 2030 (4.97 million). In the U.S., the number of diagnosed diabetes cases is projected to increase 165%, from 11 million in 2000 (prevalence of 4.0%) to 29 million in 2050 (prevalence of 7.2%). The number of women >75 years of age with diabetes will rise from >1.2 million in 2000 to >4.4 million in 2050 (+271%). The number of men with diabetes in this same age-group was projected to rise from >0.8 million in 2000 to >4.2 million in 2050 (+437%) [12]. These predictions are similar to our estimations. Papier et al. [4] performed a longitudinal study of distance learning, Open University students in Thailand estimated the 8-year cumulative incidence rate of diabetes type 2 (2005 to 2013). They found the cumulative incidence of DM were 177 per 10,000, however, this study found about 330 per 10,000 persons per year. These figures might not be comparable since the previous study was performed only in a specific subgroup of a university setting as well as only diagnosed cases were reported then. Here we used the model to estimate both diagnosed and undiagnosed incidence cases in the Thai population. Aekplakorn et al. [36] estimated the undiagnosed diabetes percentage from the national examination survey which fall within our estimates, i.e., 35.4%, while we estimated the overall percentage of undiagnosed diabetes among males being 25%, except those aged between 1 and 40 with 39%. Flores-Le Roux et al. [37] found that undiagnosed diabetes increased mortality similar to our findings (2 vs. 10 times), while our results showed different age classes had different mortality rates and were higher than previous studies. In addition, our findings on screening programs were interesting, the screening rate among females aged 15–34 years old was very high in 2005 and was reduced thereafter. This could be explained by loss to follow-up screening in the older ages, as shown in this previous study [38]. Disparity in positive screening rates between males and females in the 15–34 years age category in particular could be explained by lower accessibility to early detection and management in men, particularly in rural areas [36]. One study showed females and males also differed in health care utilization, with males generally utilizing fewer health services [39]. Deerochanawong et al. [14] pointed out that increasing screening among the high-risk population and setting a national DM screening strategy would be one of the key challenges for the management of diabetes in Thailand. There were some examples of successful sub-national diabetes screening [40]. The World Health Organization (WHO) recommended that people should be screened at 3-year intervals beginning at age 45 [41]. While, the Department of Disease Control Thailand expanded the strategic plans to screen diabetes to 90% among those aged 15 year or over [16]. Our finding showed the screening coverage among those ages were lower than the policy recommended. Our results suggested that the coverage of screening among high-risk populations (males aged between 15 and 34 years old) should be increased. Improving the data quality and the timeliness of reporting to reflect the real medical and economic burden of diabetes would benefit any planning of health policy at the national level, according to WHO recommendations which suggested actions for strengthening diabetes surveillance reporting and be kept up to date [42]. The Bureau of Epidemiology (BoE) collaborated with The Office of Disease Prevention and Control Thailand, they aimed to obtain the surveillance reports from all health service units to 80% [16]. Our model showed that overall this level of reporting might have been achieved and must be maintained. The population dynamic model of diabetes deepens on our understanding of diabetes epidemiology by accounting for the complex interactions between the population demographic changes, disease progression, and consequences of current health interventions such as screening and reporting. This model is appropriate for long-term prediction of diabetes burdens and the potential impacts of improving screening and reporting of diabetes policies. Using the Bayesian approach to model fitting was to ensure that the correlations between the model parameters were fully accounted for, and that future projections included all the uncertainties associated with the model fitting.

Our model has some limitations. For model simplicity we assumed the following, the age specific case fatality and incidence rates were constant over time, while some studies showed that case fatality rates decreased because of increasing access to hospitals [43]. Lifestyle changes might also effect the incidence rate [23]. Based on the 10-year interval population census data, the mid-year population and the constant age specific fertility and mortality were used for each 10-year interval in the population dynamic model. We assumed that the estimates of reporting among both males and females were the same. The DM data used in the model included both type 1 and type 2 diabetes for all provinces around the country except Bangkok, where the data was not obtained because of the complexity of the reporting system, especially among private hospitals in Bangkok. For future studies, it would be useful to work on the economic analysis of different screening schemes and improve the existing surveillance system. Combining the dynamic model of diabetes with cost and budget impact analysis will be a powerful tool for planning public health spending on controlling diabetes.

## 5. Conclusions

Population dynamics, together with DM incidence rates, screening and reporting rates are important components of predictions of DM prevalence. The increase in diabetes prevalence is partly attributable to demographic changes, i.e., people who live longer bare the greater chance to develop diabetes, thus contributing to the prevalence of diabetes in the long term. Our model approach gave similar findings to some previous work of DM prediction while the methodology was different [23,44]. The key findings from our study are (i) the model predicts the trend of DM prevalence will increase, especially among elderly and female, (ii) undiagnosed DM among males aged 0 to 39 is the majority among the population, (iii) mortality among undiagnosed is estimated to be 10 times higher than those diagnosed, (iv) positive screening rates have reduced among females aged 15–34 years old over the past 10 years, and, finally, (v) reporting has been estimated to lie between 84–88% over the past 10 years. This study can help to guide the health policy and decision-making at the national level by predicting the impacts of improving screening and reporting of diabetes. Further studies including economic evaluation, budget impact analysis and feasibility based on the model results and recommendations should be investigated.

We anticipate that the modeling methods described here could be used by other countries, especially those with reliable census estimates to estimate future diabetes burden, as well as the potential effects of screening and reporting. In addition, the modelling approach can be adopted to study other chronic diseases and behavior changes when demographic factors are important drivers of a population’s health burden. 

## Figures and Tables

**Figure 1 ijerph-16-02207-f001:**
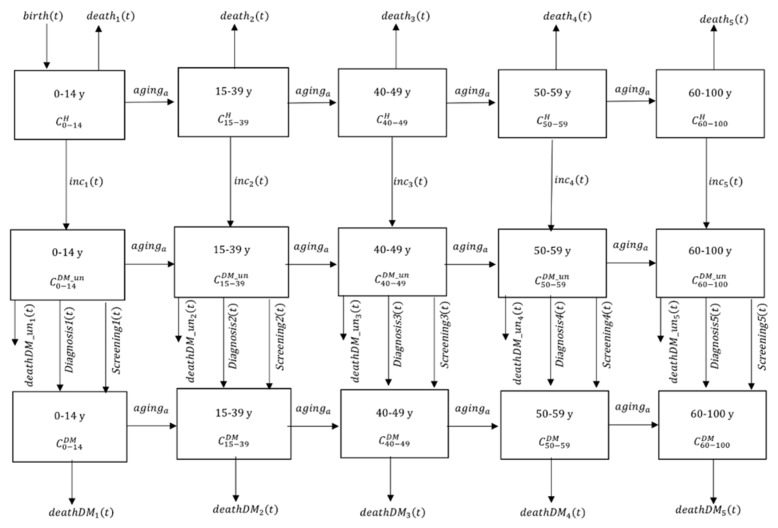
Schematic representation of the diabetes dynamic sub-model. $C_{a}^{H}$,$C_{a}^{DM\_un}$, and $C_{a}^{DM}$ denote nondiabetic, undiagnosed, and diagnosed diabetes individuals, respectively.

**Figure 2 ijerph-16-02207-f002:**
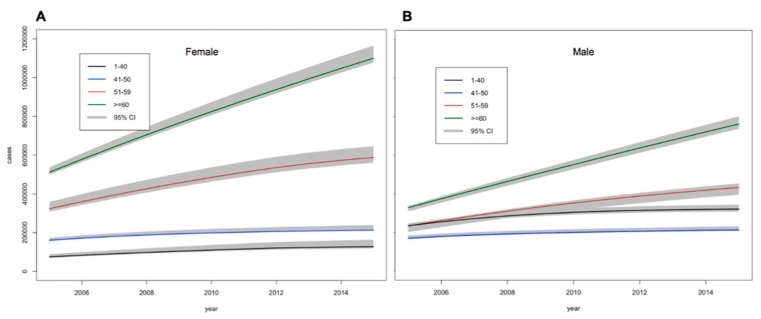
Estimations of the number (95% credible intervals) of (**A**) females and (**B**) males with diabetes between 2005 and 2015 by age.

**Figure 3 ijerph-16-02207-f003:**
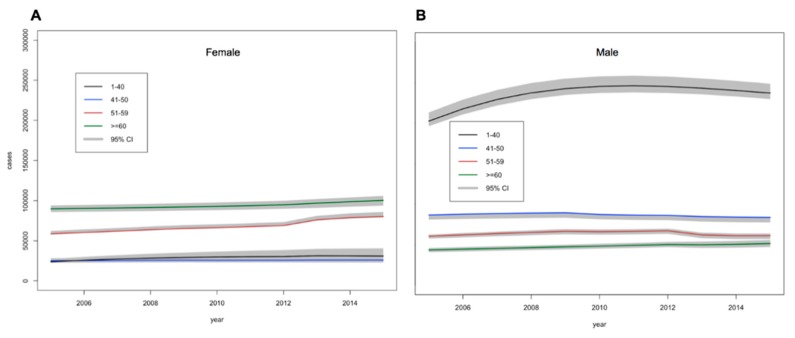
Estimations of the number (95% credible intervals) of (**A**) females and (**B**) males with undiagnosed diabetes between 2005 and 2015 by age.

**Table 1 ijerph-16-02207-t001:** Projection (in thousands) of the number of males and females with diabetes (prevalence) by age group for selected year.

Year	Age-Group (Years)								Total
	0–39		40–49		50–59		≥60		
	Male	Female	Male	Female	Male	Female	Male	Female	
2005	240	40	80	110	120	210	210	340	1350
2010	230	70	180	160	210	320	300	510	1980
2015	320	130	240	210	420	590	710	1100	3720
2035	230	100	210	210	460	620	1300	1800	4930

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
