# Peer review of "A Population Dynamic Model to Assess the Diabetes Screening and Reporting Programs and Project the Burden of Undiagnosed Diabetes in Thailand"

_ijerph, 2019, doi:10.3390/ijerph16122207_

Round 1

Reviewer 1 Report

The authors present and interesting topic here on A Population Dynamic Model to Assess the Diabetes Screening and Reporting Programs and Project the Burden of Undiagnosed Diabetes in Thailand. A well conducted research. Issues which will need to be addressed are:

Abstract look ok.

Introduction: Line 42-43 please update the data regarding prevalence of DM as per the latest IDF DIABETES ATLAS Eighth edition 2017.

In Line 50 at the end of the sentence please add a sentence regarding the acute and chronic complications of diabetes. Please include “Acute complications of diabetes include diabetic ketoacidosis (DKA), Euglycemic DKA, hyperosmolar hyperglycemic state and hypoglycemia. [Please add the citations Pubmed ID: 27156051 and 28924481]. Chronic complications of diabetes include microvascular complications such as neuropathy, retinopathy and nephropathy, and macrovascular complications such as coronary artery disease, peripheral arterial disease, and stroke. [Please add the citations Pubmed ID: 27156051 and  30816322]

Line 83-84 the sentence “In Thailand, the total diabetes population has previously been predicted to be approximately 5.4 million people in 2030” please rewrite the sentence. You are using a past tense to describe future predictions.

In methods section please explain in confounders to the study and how they were resolved.

Discussion section:

Line 225-226 sentence “specific case fatality and incidence rate was constant over time but not age, while some study that case fatality ..” is unclear and need to be rewritten.

In conclusion:

Line 238-239 sentence “Population dynamics, together with DM incidence rates, screening rates and proportions of reported are important components of predictions of DM prevalence” is unclear. What proportions of reported? Is being mentioned here.

Line 241: Authors mention “those with diabetes are more likely to live longer” … I do not agree with this statement and this needs to be rewritten.

Please explain in conclusion in a sentence how this study is going to help diabetes in Thailand.

English and grammar needs to be thoroughly checked throughout the manuscript

Author Response

Response to Reviewer 1 Comments

Point 1: The authors present and interesting topic here on A Population Dynamic Model to Assess the Diabetes Screening and Reporting Programs and Project the Burden of Undiagnosed Diabetes in Thailand. A well conducted research. Issues which will need to be addressed are:

Abstract look ok.

Introduction: Line 42-43 please update the data regarding prevalence of DM as per the latest IDF DIABETES ATLAS Eighth edition 2017.

Response 1: We thank the reviewer for the useful advice. We have updated the data and changed some text as follow:

“The prevalence of Diabetes Mellitus (DM) including type 1, type 2, and gestational diabetes is increasing globally and predicted to rise from 425 million adult cases in 2017 to 629 million in 2045 [1].” - Page 1, Paragraph 1, Line 42-43.

Point 2 : In Line 50 at the end of the sentence please add a sentence regarding the acute and chronic complications of diabetes. Please include “Acute complications of diabetes include diabetic ketoacidosis (DKA), Euglycemic DKA, hyperosmolar hyperglycemic state and hypoglycemia. [Please add the citations Pubmed ID: 27156051 and 28924481]. Chronic complications of diabetes include microvascular complications such as neuropathy, retinopathy and nephropathy, and macrovascular complications such as coronary artery disease, peripheral arterial disease, and stroke. [Please add the citations Pubmed ID: 27156051 and  30816322]

Response 2: We agree with the reviewer and have added the following sentence:

“Acute complications of diabetes include diabetic ketoacidosis (DKA), Euglycemic DKA, hyperosmolar hyperglycemic state and hypoglycemia [2, 3]. Chronic complications of diabetes include microvascular complications such as neuropathy, retinopathy and nephropathy, and macrovascular complications such as coronary artery disease, peripheral arterial disease, and stroke [3, 4].”- Page 2, Paragraph 1, Line 54-58.

Point 3 : Line 83-84 the sentence “In Thailand, the total diabetes population has previously been predicted to be approximately 5.4 million people in 2030” please rewrite the sentence. You are using a past tense to describe future predictions.

Response 3: We agreed with the reviewer and have corrected the following sentence:

“It was predicted previously that the total diabetes population in Thailand would be approximately 5.4 million people in 2030” - Page 2, Paragraph 5, Line 91-92.

Point 4 : In methods section please explain in confounders to the study and how they were resolved.

Response 4: The reviewer has rightly pointed out that there could be some confounders in this study. Behaviour changes including lifestyle, dietary and exercises could influence the incidence of diabetes. Being a modelling study, we decided to state this point in the model assumptions.

“It is possible that the incidence of diabetes could be influenced by some possible confounders such as behaviour changes including lifestyle, dietary and exercises for example [5, 6]. For the study simplicity, these factors were not included in the modelling.” - Page 3, Paragraph 3, Line 133-136.

Discussion section:

Point 5 : Line 225-226 sentence “specific case fatality and incidence rate was constant over time but not age, while some study that case fatality ..” is unclear and need to be rewritten.

Response 5: We agreed with the reviewer and have corrected the following sentence:

“the age specific case fatality and incidence rates were constant over time, while some study showed that case fatality decreased because of increasing accessing to hospital [38].”

 - Page 6, Paragraph 2, Line 255-256.

In conclusion:

Point 6 : Line 238-239 sentence “Population dynamics, together with DM incidence rates, screening rates and proportions of reported are important components of predictions of DM prevalence” is unclear. What proportions of reported? Is being mentioned here.

Response 6: We corrected the sentence to:

“Population dynamics, together with DM incidence rates, screening and reporting rates are important components of predictions of DM prevalence.” - Page 7, Paragraph 2, Line 282-283. 

Point 7 : Line 241: Authors mention “those with diabetes are more likely to live longer” … I do not agree with this statement and this needs to be rewritten.

Response 7: We agreed with the reviewer and have corrected the following sentence:

“people who live longer bare the greater chance to develop diabetes thus contributes to the prevalence of diabetes in a long term.” - Page 6, Paragraph 3, Line 241.

Point 8 : Please explain in conclusion in a sentence how this study is going to help diabetes in Thailand.

Response 8: We thank the reviewer for the useful advice. and have added the following sentence:

“This study can help guiding the health policy and decision-making at the national level by predicting the impacts of improving screening and reporting of diabetes for example. Further studies including economic evaluation, budget impact analysis and feasibility based on the model results and recommendations should be investigated.” - Page 7, Paragraph 1, Line 292-295.

Point 9 : English and grammar needs to be thoroughly checked throughout the manuscript

Response 9: We have thoroughly checked our manuscript for language usage, spelling, and grammar as advice.

Reference

1.           IDF IDF Diabetes Atlas. http://www.diabetesatlas.org (27 May 2019),

2.           Rawla, P.; Vellipuram, A. R.; Bandaru, S. S.; Pradeep Raj, J., Euglycemic diabetic ketoacidosis: a diagnostic and therapeutic dilemma. Endocrinol Diabetes Metab Case Rep 2017, 2017.

3.           Gregg, E. W.; Sattar, N.; Ali, M. K., The changing face of diabetes complications. Lancet Diabetes Endocrinol 2016, 4, (6), 537-47.

4.           Li, J.; Cao, Y.; Liu, W.; Wang, Q.; Qian, Y.; Lu, P., Correlations among Diabetic Microvascular Complications: A Systematic Review and Meta-analysis. Sci Rep 2019, 9, (1), 3137.

5.           Feldman, A. L.; Long, G. H.; Johansson, I.; Weinehall, L.; Fharm, E.; Wennberg, P.; Norberg, M.; Griffin, S. J.; Rolandsson, O., Change in lifestyle behaviors and diabetes risk: evidence from a population-based cohort study with 10 year follow-up. Int J Behav Nutr Phys Act 2017, 14, (1), 39.

6.           Green, A. J.; Bazata, D. D.; Fox, K. M.; Grandy, S.; Group, S. S., Health-related behaviours of people with diabetes and those with cardiometabolic risk factors: results from SHIELD. Int J Clin Pract 2007, 61, (11), 1791-7.

Reviewer 2 Report

Comments 

General:This manuscript performs a study that aims to use a population dynamic model overlaid with a diabetes dynamic sub-model to predict the disease burden and in particular the mortality of undiagnosed diabetes in the Thai population by age and sex, and to assess the diabetes screening program and  reporting system. The authors used a Bayesian framework which permits the inclusion of uncertainty in onset and reporting rates.

The authors present very good information about diabetes, a pathology that in this country affects a very high percentage of people with a very early onset compared to other populations. This type of study is particularly relevant in analyzing and improving diabetes detection and control policies, as the authors mention throughout the article.

Manuscript title: the title is correct

Abstract:  It is correct

Introduction: The literature used is pertinent to the study and the purpose of the study was clearly stated. 

Methods: the study was clearly stated. 

Results: the choice of tables and figures are correct, clearly summarize what is discussed later.

Discussion and Conclusions: I suggest that the authors identify more clearly the strengths of the study.

References: There were 39 and all appropriate.

Author Response

Response to Reviewer 2 Comments

General:This manuscript performs a study that aims to use a population dynamic model overlaid with a diabetes dynamic sub-model to predict the disease burden and in particular the mortality of undiagnosed diabetes in the Thai population by age and sex, and to assess the diabetes screening program and  reporting system. The authors used a Bayesian framework which permits the inclusion of uncertainty in onset and reporting rates.

The authors present very good information about diabetes, a pathology that in this country affects a very high percentage of people with a very early onset compared to other populations. This type of study is particularly relevant in analyzing and improving diabetes detection and control policies, as the authors mention throughout the article.

Manuscript title: the title is correct

Abstract:  It is correct

Introduction: The literature used is pertinent to the study and the purpose of the study was clearly stated.

Methods: the study was clearly stated.

Results: the choice of tables and figures are correct, clearly summarize what is discussed later.

Point 1: Discussion and Conclusions: I suggest that the authors identify more clearly the strengths of the study.

Response 1: We thank the reviewer for the useful advice. We have added the strength of the model in the discussion as follow:

“The population dynamic model of diabetes deepens our understanding of diabetes epidemiology by accounting for the complex interactions between the population demographic changes, disease progression and consequences of current health interventions such as screening and reporting. This model is appropriate for long-term prediction of diabetes burden and the potential impacts of improving screening and reporting of diabetes policies. Using Bayesian approach to model fitting was to ensure that the correlations between the model parameters were fully accounted for, and that future projections included all the uncertainties associated with the model fitting.” - Page 6, Paragraph 1, Line 246-253.

References: There were 39 and all appropriate.